# Off-Axis Diffractive Optics for Compact Terahertz Detection Setup

**Paweł Komorowski** [1,*], **Mateusz Surma** [2], **Michał Walczakowski** [3],
**Przemysław Zagrajek** [3] **and Agnieszka Siemion** [2,*]

1   Institute of Microelectronics and Optoelectronics, Warsaw University of Technology, Koszykowa 75, 00662 Warsaw, Poland
2   Faculty of Physics, Warsaw University of Technology, Koszykowa 75, 00662 Warsaw, Poland; mateusz.surma.dokt@pw.edu.pl
3   Institute of Optoelectronics, Military University of Technology, Kaliskiego 2, 00908 Warsaw, Poland; mwalczakowski@wat.edu.pl (M.W.); przemyslaw.zagrajek@wat.edu.pl (P.Z.)
*   Correspondence: p.komorowski@imio.pw.edu.pl (P.K.); agnieszka.siemion@pw.edu.pl (A.S.)

**Abstract:** Medical and many other applications require small-volume setups enabling terahertz imaging. Therefore, we aim to develop a device for the in-reflection examination of the samples. Thus, in this article, we focus on the diffractive elements for efficient redirection and focusing of the THz radiation. A terahertz diffractive optical structure has been designed, optimized, manufactured (using extrusion-based 3D printing) and tested. Two manufacturing methods have been used—direct printing of the structures from PA12, and casting of the paraffin structures out of 3D-printed molds. Also, the limitations of the off-axis focusing have been discussed. To increase the efficiency, an iterative algorithm has been proposed that optimizes off-axis structures to focus the radiation into small focal spots located far from the optical axis, at an angle of more than 30 degrees. Moreover, the application of higher-order kinoform structure design allowed the maintaining of the smallest details of the manufactured optical element, using 3D printing techniques.

**Keywords:** THz; diffractive optics; 3D printing; skin cancer diagnosis; iterative design; off-axis structures; beam redirecting

## 1. Introduction

The terahertz radiation spectrum has been at the center of researchers' attention for the last few decades [1] due to its specific features. These features comprise non-invasiveness [2–4] and non-ionization as well as high absorption and dispersion in water [5] which is a major component of biological tissues. THz radiation is starting to play an increasingly significant role in many biomedical applications [6] including detection of e.g., skin cancer [7,8]. Our ambition is to design, manufacture and verify thin diffractive optical elements (DOEs) which would allow the development of compact and more applicable THz skin scanners. Using off-axis DOEs is advantageous due to the possibility of sustaining a small size of the focal point while significantly reducing a minimal focusing distance (unlike on-axis diffractive lenses and refractive lenses [9]). In the recent publication [10] we have already presented in-reflection inspection setup with plane wave-to-point "emitter lens" and point-to-point "detector lens". The whole setup has been designed for the frequency $\nu$ = 520 GHz. The choice is connected with the distinguishability of healthy and tumorous skin tissues, which is the best in the 0.35–0.55 THz frequency range [11,12]. Further work aims at optimization of inspection setup by applying point-to-point lenses on both sides and reducing focal lengths further. In this article the theoretical and experimental comparison of off-axis diffractive structures for the THz spectrum

is discussed. For the design wavelength (520 GHz), different 3D printing methods were used to manufacture structures. First, more advanced powder-based technique [13] enabled the creation of simple structures [14,15], but not all small features of the structure could be properly resolved in the case of higher frequencies. Next, a much simpler and more cost-efficient extrusion-based method and a different optical design have been used. The reduction of the focal length is a key feature to setup minimization. At the same time, the diameter of the structure should remain as large as possible to ensure sufficient signal intensity. These two assumptions lead to the off-axis lenses with small numerical apertures (*NA* which is also related to $\frac{f}{d}$). The shorter the focal lengths, the stronger the influence of the aberrations on the focal spot, especially for the off-axis focusing. These effects can be partially reduced with the proposed iterative design algorithm. Moreover, for the frequency 520 GHz the attenuation of optical materials starts to play significant role. Thus, the application of paraffin is proposed. It has very low attenuation coefficient even up to 1 THz, while most of the 3D printable materials are characterized by higher values of absorption. To be able to manufacture such paraffin lenses, another way of coding phase delay maps has been chosen—high-order kinoform (HOK).

## 2. Design and Modeling

The concept of a compact in-reflection setup for skin cancer diagnosis is presented in Figure 1. The two designed diffractive optical elements are working in transmission and additionally are redirecting the incident radiation off-axis and focus the beam at a particular distance.

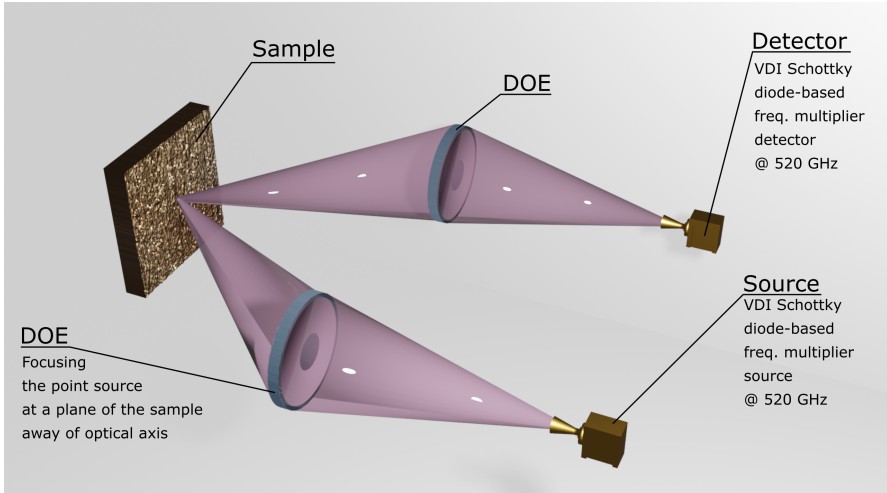

**Figure 1.** The configuration of the in-reflection setup for the sample examination using two off-axis diffractive structures (DOEs). The radiation from the divergent source illuminates the diffractive element focusing it at some angle and at some distance after the structure. Such solution can help in minimizing the distance between the element and the sample, thus the off-axis focusing is required. Then, the radiation reflected or scattered on the sample is collected by the element with the same shape and geometry to focus this radiation on the detector. The concept is general, allowing for the arbitrary choice of the source and detector of radiation as well as sample under test. Here, VDI Schottky diodes with frequency multipliers have been used.

In this approach, two point-to-point lenses are used (marked as DOE). Due to the symmetry of the setup, both lenses can actually be considered to be the same diffractive structure, rotated by 180°. The structure needs to be thin and light with short focal length, which allows focusing of THz radiation into a small focal spot located outside the contours of the device. These restrictions practically exclude the application of refractive lenses; as such, lenses would be too bulky and heavy [9]. Using a refractive lens in the described geometry would require either partial illumination of a larger lens or mounting the lens at some angle to the structure [16]. In the first case, the f-number of the lens would be very small (below 1). Such lenses are extremely thick and challenging to manufacture as refractive

structures. In the second case, the focal spot would be very elongated (a sample would be placed at an angle to the focal plane of the lens). On the other hand, redirection of the terahertz radiation with use of mirrors is also possible and commonly used [17]. However, once again, in the short-focal-length and off-axis focusing design, mirrors would result in unnecessary bulkiness of the system. The redirection and focusing of the radiation with parabolic mirrors at an angle higher than 90° is also challenging. Therefore, our research has been focused on diffractive structures with thickness of the order of several millimeters. Light propagation in the Fresnel region and modified convolution method have been used in numerical simulations [18]. All calculations have been performed on 4096 × 4096 pixels square matrix with 117 μm sampling. The design frequency was equal to 520 GHz which corresponds to 0.58 mm design wavelength. Three different design approaches were verified: cutting only part of off-axis lens described by analytical equation, the back-propagation of diffraction-limited focal spot, and the iterative algorithm. In the first method, the diffractive lens is designed according to the lens equation in the non-paraxial approach, and its off-axis part is taken. The second method relies on the definition of the target intensity distribution (ideal focal spot) and propagating it at the negative distance to the lens plane. The third approach, in the simplified view, is based on the iterative implementation of previous two methods. The phase distribution of the structure obtained using the first method can be seen in the top left part of Figure 2.

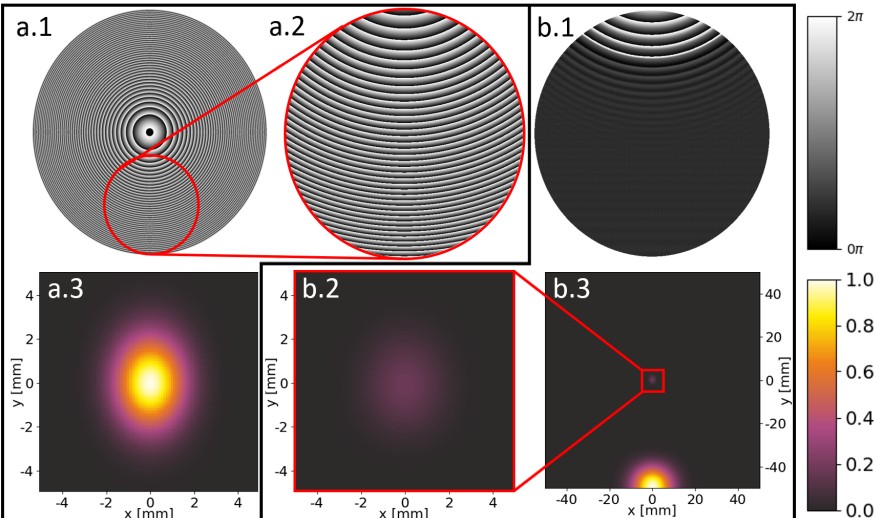

**Figure 2.** Two concepts of designing off-axis lens—analytical equation (a) and re-propagation of ideal focal spot (b). Subimages (**a.1,a.2**) show the phase distributions of the whole lens designed with the analytical equation in the non-paraxial approach and its part taken for image reconstruction, respectively. The theoretical focal spot obtained from the cut-out lens can be seen in (**a.3**). The phase distribution obtained after back-propagation of the predefined focal spot is shown in the subfigure (**b.1**) (assuming also Gaussian shape illumination resulting from the divergence of the used radiation source). The image reconstructed using this method is presented in (**b.2**). The same focal plane can be seen in wider scale in the subimage (**b.3**).

A diffractive off-axis lens coded in the form of kinoform with the focal length of 100 mm has been designed. The phase distribution is described by the following equation after applying modulo $2\pi$ operation:

$$\phi(r) = -\frac{2\pi}{\lambda}\sqrt{f^2 + r^2},\tag{1}$$

where $\phi$ is the phase delay introduced by the lens, $r$ is radial coordinate (calculated from the main optical axis), $\lambda$ is the design wavelength and $f$ is the focal length. Next the side part of this lens, placed 60 mm from the center and with the diameter of 75 mm, has been cut. In this way, the original focal point is located 100 mm behind the lens plane and 22.5 mm up from its edge, lying outside of the

contour of the lens. As it can be seen, this lens focuses radiation in the desired position with a good efficiency, but the spot is relatively big and elongated.

Second approach based on re-propagation of the optical field is shown in Figure 2b. The ideal focal spot (diffraction-limited intensity distribution with flat phase) is defined in the focal plane. The assumed background of the defined focal spot is not perfectly black, but in the very dark grey shade. It allows the preservation of the information about the phase distribution without affecting the result in any significant way. Then, light is propagated backward to the lens plane and a shifted circular aperture with 75 mm in diameter is cut from the obtained phase distribution (Figure 2(b.1)). It can be seen that after back-propagation at a distance of 100 mm such a defined phase distribution of light diverges only to a certain diameter. This diameter is insufficient to properly describe off-axis structure with the described above geometrical parameters. As a result, the majority of the lens area is relatively flat and only its top part is active and works as intended. Nevertheless, the focal spot with dimensions smaller than in the first case (using analytical equation) is obtained (Figure 2(b.2)). However, its intensity is much lower due to the fact that most of the energy is not redirected and forms an unwanted bright region in the middle of the structure. This non-diffracted zeroth order can be seen at the bottom of intensity distribution shown in the Figure 2(b.3)). The third approach is a combination of previous two cases. The light is propagated back and forth from the lens plane to the focal plane in an iterative algorithm (similar to Gerchberg-Saxton [19] or ping pong [20] algorithms). In the lens plane, the calculated amplitude is overwritten with the amplitude of the incident plane wave, while the amplitude of the ideal focal spot is defined in the focal plane. The phase distribution obtained in the lens plane defines the optimized off-axis lens. The algorithm converges fast and in just a few iterations the output light field distribution matches with the desired one well. The phase distributions in the lens plane and the intensity distributions in the focal plane for first three iterations are shown in Figure 3.

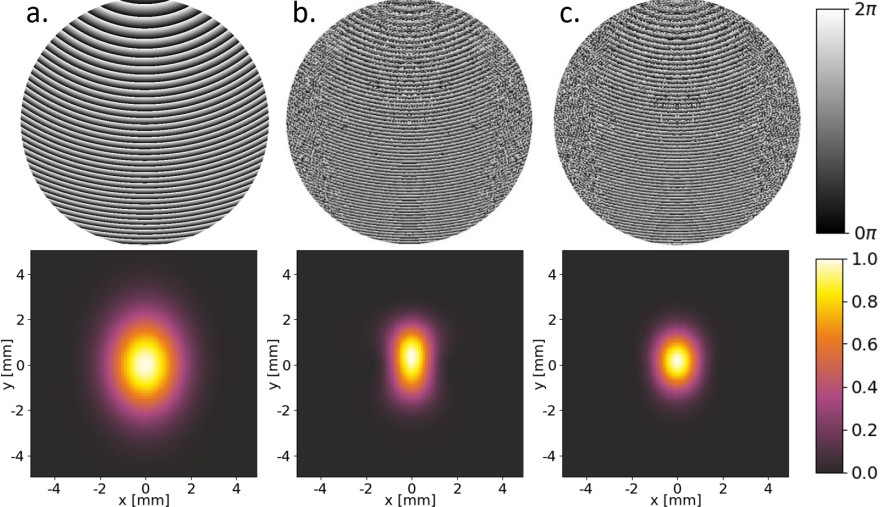

**Figure 3.** The comparison of the phase distributions (**top** panels) and focal spots (**bottom** panels) of the off-axis lens in the first three steps of the iterative algorithm for: (**a**. a single propagation, **b**. the second iteration and **c**. the third iteration).

In the first step, the light is focused by the lens created according to the analytical equation as in the first method. Then the amplitude distribution is replaced by ideal point distribution as in the second method. The amplitude was modified but the phase distribution was maintained as calculated after propagation. Next the light field was propagated back to the lens plane. The new lens is illuminated once more with initial plane wave distribution and the second iteration of algorithm begins. As it can be seen, the second focal spot is already significantly smaller than the first one, but still noticeably elongated. However, in the third step the obtained focal spot is circular and has

dimensions in order of diffraction limit. Full widths at half maximum (FWHM) of calculated focal spots are equal to 3.33 mm, 2.22 mm and 1.99 mm, respectively. These values have been obtained by averaging FWHMs for horizontal and vertical axes. The diffraction limit of light for this wavelength and focusing parameters (Airy dot diameter, described as $A_d = 2 \times 1.22 \times \frac{\lambda f}{d}$) is equal to 1.88 mm. Thus, FWHM of the focal spot obtained from the optimized off-axis lens is only 6% higher than the theoretical limit. Further iterations do not provide any significant improvements which was verified in numerical simulations.

## 3. Structures Manufacturing

The iteratively optimized diffractive elements have been fabricated from Polyamide 12 (PA12) using additive manufacturing (3D printing) by powder-based selective laser sintering (SLS) and extrusion-based method using filament. So far, the main problem with these methods has been the smallest thickness of the high wall equal to around 0.8 mm and printing resolution of around 0.05 mm (depending on a printing method and a device). Such a resolution is insufficient for the details of the sophisticated DOEs. In the case of the designed structure, the narrowest zone is around 0.82 mm thick and 0.94 mm high (with the details of the optimized structure being even smaller). It is not possible to manufacture these details using the most of 3D printing techniques. Therefore, the usage of higher-order kinoforms (HOKs) is proposed here. They are designed as classical diffractive lenses for small integer multiples of basic wavelength. As a result, these structures introduce a phase delay ranging from 0 to the multiple of $2\pi$, instead of $0 - 2\pi$ phase shift. Nevertheless, the optimization has always been performed for the basic wavelength which in this case has been a harmonic of the design wavelength. HOKs are thicker than classical diffractive lens by a factor which is equal to the order of the kinoform $p$ but their zones are also respectively wider. Several HOKs of low order (1st, 2nd and 4th order) with respectively bigger details of the structures have been designed.

## 4. Experimental Evaluation

The performance of all manufactured structures has been experimentally verified in the setup, shown in the Figure 4.

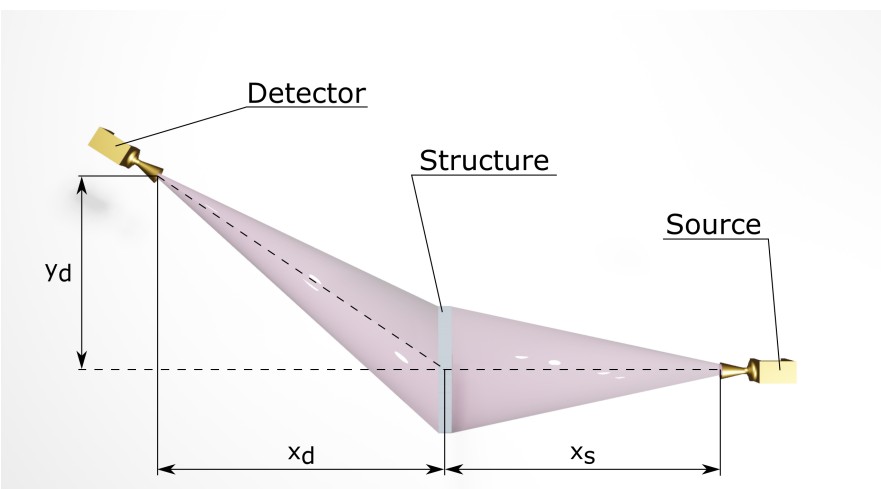

**Figure 4.** The measurement setup for off-axis diffractive lenses with frequency multipliers with Schottky diodes as a source and a detector of THz radiation. $x_s$ = 100 mm is the distance from the source of radiation to the thin structure, $x_d$ = 100 mm corresponds to the distance along optical axis from the structure to the image plane and $y_d$ = 60 mm reflects the perpendicular shift of the detector (and focal point) from the optical axis.

All marked dimensions have been chosen empirically before designing optical elements taking into account dimensions of available sources and detectors as well as total size and usability of a

future scanner head. This optical setup consists of two frequency multipliers with Schottky diodes to emit and detect THz radiation with the structure under test in the middle. The comparison of the 1st, the 2nd and the 4th order kinoforms as well as corresponding focal spots are presented in Figure 5. The scans of the analogous structures performed along the optical axis can be found in our previous paper [10]. Their character remained the same.

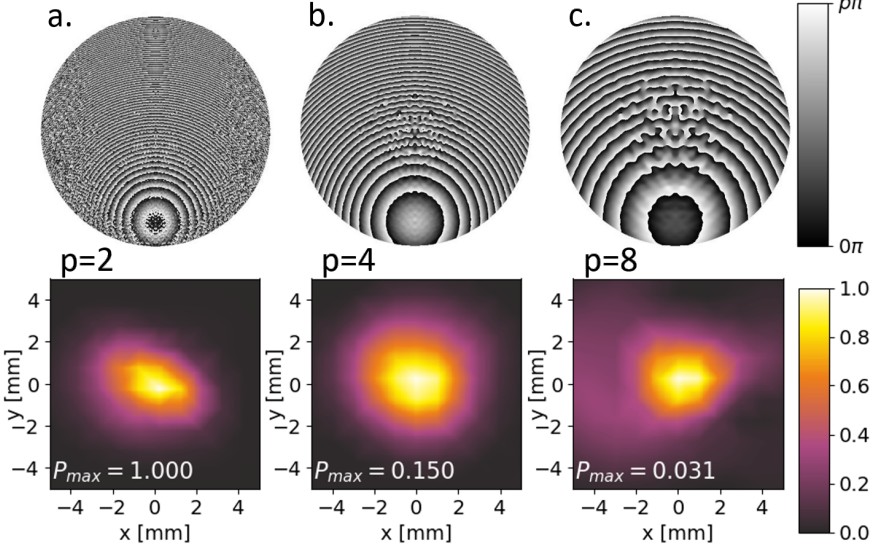

**Figure 5.** The comparison of phase delay maps (top panels) of **a**. 1st ($p = 2$), **b**. 2nd ($p = 4$) and **c**. 4th ($p = 8$) order kinoforms with normalized intensity distributions in focal planes (bottom panels). All intensity images are normalized to unity, while the maximum power detected for each structure in respect to the value registered for the first order kinoform (**a**) is denoted as $P_{max}$.

It can be noticed that in subsequent HOKs respective zones are indeed wider proportionally to their order. The designed structures generally resemble Fresnel lenses. Additional discontinuities on the borders of zones (regions where phase drops from $2\pi$ to 0) result from the iterative optimization algorithm and enhance geometric parameters of focal spots. All intensity distributions have been separately normalized for the sake of visibility but measured peak values have been denoted on images as $P_{max}$ (in relation to the best performing structure). It can be seen that for kinoforms of higher-order intensity of focal spot significantly decreases (by 85% for the 2nd order and by almost 97% for the 4th order kinoform). It is clearly connected with the thickness of the structures. The absorption coefficient of PA12 is already relatively high at 520 GHz (around 2.2 cm$^{-1}$) [21] and starts to play a significant role when the thickness of the structures is doubled or quadrupled. The transparency issue can be partially solved by application of less attenuating media. However, the number of materials accessible for 3D printing is rather limited. Therefore, another manufacturing method has been proposed based on the additive manufacturing of the molds. These molds correspond to the designed DOEs and then are filled with melted paraffin. Thus, the mold corresponding to the fourth order kinoform has been manufactured and three kinds of paraffin from different manufacturers have been tested. These kinds of paraffin have been marked as b, c and d where b refers to transparent paraffin, c to white paraffin and d to reused paraffin. The experimental results together with reference PA12 lens are shown in Figure 6.

Here, all power distributions have also been normalized separately and the ratios of recorded maxima to the peak power obtained from reference lens are denoted as $P_{max}$. The intensities of focal spots obtained for all three paraffin HOKs are significantly higher than the one from PA12 lens (by 42%, 76% and 36%, respectively). On the other hand, differences between very similar paraffin samples obtained from different distributors are also far from being negligible. Finding the reasons for that would require more in-depth study, especially of the differences between materials. Initial

measurements show that different types of waxes and paraffins have significantly different THz optical properties.

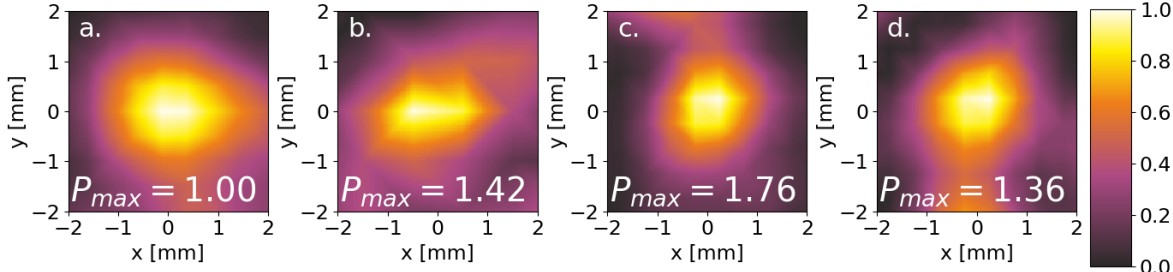

**Figure 6.** Off-axis focal spots, obtained from **a**. PA12 lens, **b**. transparent paraffin lens, **c**. white paraffin lens and **d**. reused paraffin lens. All images are normalized to unity, while the maximum power detected for each structure in respect to the PA12 lens is denoted as $P_{max}$.

## 5. Shorter Focal Lengths

As a proposed design method has proved to be successful in designing off-axis lens-like structures with the focal length equal to 100 mm, the additional simulations have been performed. These numerical calculations allowed determination of how short the focal length can be while maintaining good geometrical parameters of the focal spot. The diameters of lenses have been cropped to 50 mm in all these simulations. However, it must be noted that here as well as in previously described cases the reconstructing beam is described with flat phase and Gaussian intensity distribution. It means that the introduced 50 mm aperture has not limited active structure region significantly. The phase distributions of four lens-structures with focal lengths varying from 50 mm to 10 mm together with the corresponding focal spots are presented in Figure 7.

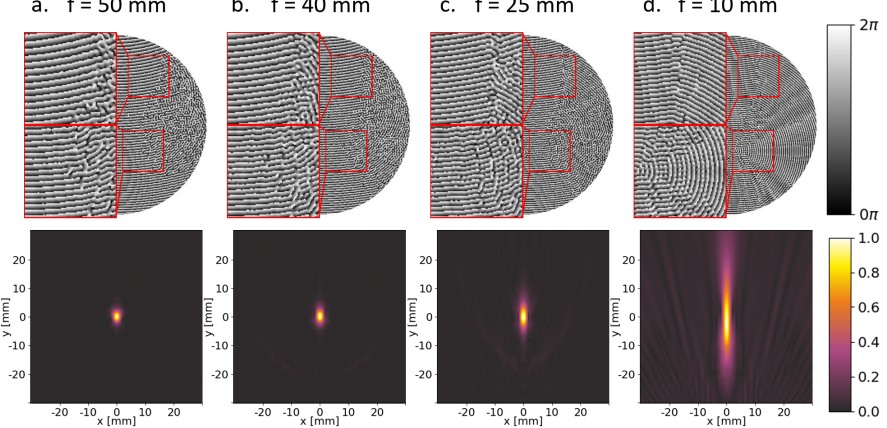

**Figure 7.** The comparison of phase distributions (top panels) and calculated focal spots (bottom panels) of the off-axis lenses with varying focal lengths: **a**. f = 50 mm, **b**. f = 40 mm, **c**. f = 25 mm and **d**. f = 10 mm. The same two parts of all lenses have been magnified to make fine changes in the phase maps more visible.

The longest focal length chosen to be shown in this figure is 50 mm, as no significant changes can be observed in the intensity distribution for longer focal lengths (like the 100 mm focal spot shown in Figures 3 and 5). On the other hand, distortions from the circularity of the focal spot become more and more visible for smaller focal lengths. For $f = 25$ mm the focal spot is already significantly elongated, while for $f = 10$ mm it resembles a stripe not a dot.

The additional study on the shape of focal spots is presented in Figure 8.

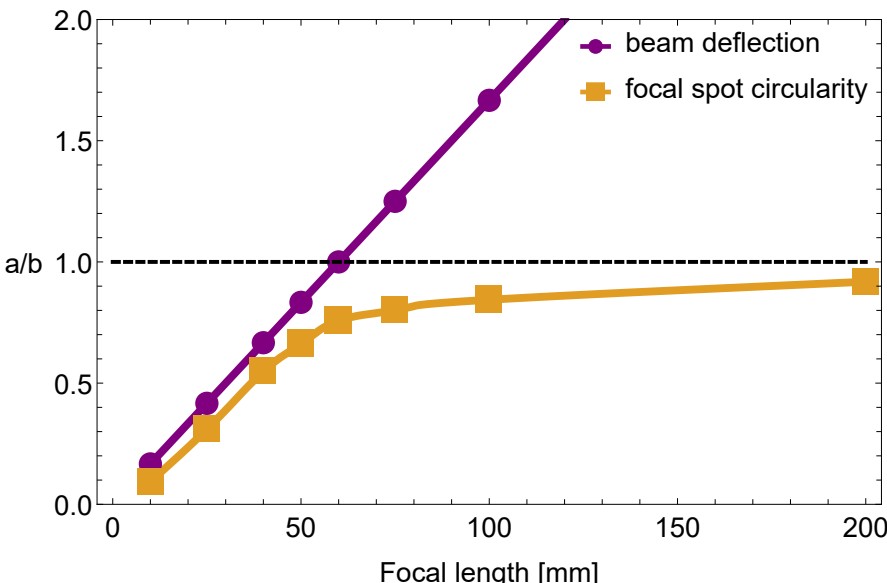

**Figure 8.** The dependence of the circularity of the focal spots (orange line) and the beam deflection (purple line) on the focal length of the designed lens. The circularity of the spot is defined as the ratio of its diameters measured along horizontal and vertical axes (a/b). The beam deflection denotes the tangent of the angle between the lens surface and the deflected beam.

The orange line denotes circularity of the focal spots, calculated as the ratio of the diameters of the spots measured along horizontal (a) and vertical (b) axes. The violet line corresponds to the degree of the deflection of the incident beam—in this case it is simply a tangent of the angle between the lens surface and the deflected beam. The circularity equal to one would denote the perfect circle, but the smaller focal length the closer it gets to zero. It means that the shape is more elliptical. The dependence of the spot circularity on the focal length is in the shape of two lines with different slopes, crossing around focal length which equals to 60 mm. As it can be seen in the Figure 8, for longer focal lengths the dependence is almost flat while for shorter ones the curve is almost parallel to the beam deflection line (up to the value of circularity equal to 1). Mathematically, this curve can be described by two asymptotes, as in Equation (2):

$$\begin{cases} \frac{a}{b} = 1 & \text{for } f \to \infty, \\ \frac{a}{b} = N = \frac{f}{y_d} & \text{for } f \to 0. \end{cases} \tag{2}$$

The long-focal-length-limit behavior of the curve is easily understandable, as the shift from the main optical axis to the center of the structure becomes much smaller than the focal length. In this case the structure starts to neglect off-axis requirements and it is focusing almost on-axis with the perfect circular focal spot. The interesting behavior is observed for a short-focal-length-limit as the curve approaches tangent of the deflection angle of the THz beam. It seems to be the fundamental limit of off-axis focal spot circularity, at least in this design method. The breakthrough point lies at 60 mm, where the focal length is equal to the shift from the optical axis and two asymptotes intersect.

## 6. Summary

The off-axis diffractive lens-like structures for THz spectral range have been designed, manufactured and tested. The applicability of the higher-order kinoforms has been shown. They offer a better manufacturing quality at the expense of the increased losses introduced by the structure thickness. The further work aims at solving this issue by changing the used PA12 material for the one with the lower absorption coefficient (paraffin). The paraffin lenses manufactured by filling 3D-printed molds are currently considered to be the most promising option and have shown satisfactory results.

Additional study on the shape of focal spots circularity for different focal lengths has been conducted. Summarizing, the changing point between off and on-axis approaches can be set for the focal length of the structure equal to the shift of the structure from the main optical axis.

All simulations and experimental evaluation have allowed determination that the smallest focal length of structures is equal to 50 mm for the configuration of the optical setup proposed here for in-reflection skin cancer scanners. At the same time, it needs to be remembered that in the proposed setup the middle of the structure is shifted 60 mm from the main optical axis.

The analysis of the structures allowed determination of to what extend the setup can be minimized. The off-axis focusing is challenging, as tremendous aberrations are observed. The beam deflection angle for the proposed lens is equal to almost 31°. For the shortest focal length with acceptable distortions of the focal spot ($f$ = 50 mm) the mentioned angle even reaches 51°. It is well beyond the limits of paraxial approximation (which can be applied up to approx. 17–18°). The iterative algorithms proposed in this article allow significant reduction of the undesired aberration effects and obtain higher quality of the off-axis focal spots.

**Author Contributions:** Conceptualization, P.K., A.S.; methodology, P.K., A.S., software, P.K.; investigation, M.W., P.Z.; writing—original draft, P.K.; writing—review and editing, M.S., A.S.; visualization, M.S., P.K.; supervision and idea, A.S.; project administration, A.S; funding acquisition, A.S. All authors have read and agreed to the published version of the manuscript.

**Funding:** Research funded by the National Center for Research and Development under the LIDER program (LIDER /11/0036 / L-9/17 / NCBR / 2018).

**Acknowledgments:** Authors would like to thank Orteh Company for providing LS 6.0 software for execution of light propagation simulations according to scalar diffraction theory.

**Conflicts of Interest:** The authors declare no conflict of interest. The funders had no role in the design of the study; in the collection, analyses, or interpretation of data; in the writing of the manuscript, or in the decision to publish the results.

## Abbreviations

The following abbreviations are used in this manuscript:

DOE     Diffractive optical element
HOK     Higher-order kinoform
FWHM     Full width at half maximum
PA12     Polyamide 12

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

**Sample Availability:** Additional manufacturing data (including precise phase distributions of the structures) can be obtained from the corresponding author upon reasonable request.

