# Peer review of "Off-Axis Diffractive Optics for Compact Terahertz Detection Setup"

_applsci, doi:10.3390/app10238594_

Round 1

Reviewer 1 Report

  1. In the introduction authors say that the frequency for which DOE are designed is 520 GHz. Why this specific frequency was chosen? Are there any indications that it is superior/most suitable to skin cancer cell  in the skin? I think this should be clarified.
  2. The description of Fig.2 does not accurately reflect what is actually depicted there, therefore more detailed explanation is needed. Also, axis scales in bottom panels are needed.
  3. In line 52 authors claim that "...spot is relatively big and elongated". However, it is hard to perceive the dimensions of the spot. This problem would be resolved adding the axis scale in Fig. 2, as mentioned in my remark No. 2. Also, how does the focusing properties of off-axis DOE compare with focusing properties of conventional off-axis parabolic mirror?
  4. Lines 54-56 state that "Then, light is propagated backward to the lens plane and a shifted circular aperture with 75 mm in diameter is cut from the obtained phase distribution." I wonder how can be the phase distribution obtained from light backpropagation from focus plane to the lens plane? I think a brief explanation in the text or at least a reference would be useful for the general understanding.
  5. Line 60: "The focal spot with dimensions smaller than in the first case (using analytical equation) is obtained."

    As I understand the focal spot size was already defined and taken as the parameter for use in backpropagation method, so in my opinion this sentence is rudimental, as it does not carry any new information. Again, the scale in Fig.2 would help a lot!

  6. Line 73: " Then the amplitude distribution is replaced by ideal point distribution as in the second method."

    Comparing Fig.2 and Fig.3 it looks like the amplitude in this case is much higher than the one defined for the use in pure backpropagation method. For me it looks like it is enough to take the amplitude intensity defined by the analytical method shown in Fig 2. then take the diffraction limited spot size, combine them together and have the input for the iterative algorithm. This would imply that there is no use to show/discuss the results of backpropagation at all.

  7. Line 79: "Full widths at half maximum (FWHM) of calculated focal spots are equal to 3.33 mm, 2.22 mm and 1.99 mm, respectively." Along which direction these numbers were obtained? Axis scales to Fig 3. panels showing focus points are needed.
  8. Line 107: " The comparison of the 1 st, the 2nd and the 4th order kinoforms as well as corresponding focal spots are presented in Fig. 5."

    How do authors know that they are the focal plane? The beam profile along optical axis of the setup needs to be provided in order to show where the focus of the lens is and to prove that the measurements are done really in the focal plane.

  9. It would help if phase delay order would be marked in Fig. 5 for each DOE design, also spaces are needed in axis labels between the quantity and units of measurement.
  10. Line 125: " The experimental results together with reference PA12 lens are shown in Fig. 6" Here the focused beam intensity along z-axis needs to be shown also.
  11. In Fig. 6 the space is needed between the quantity and units of measurements in axis labels.
  12. Fig. 7: XY scales are needed in panels demonstrating the focal plane.

Reviewer 2 Report

Komorowski et al present a work based on the development of off-axis diffractive optics for THz devices, comparing theoretical and experimental results. They develop 3D-printing for the lenses reducing focal lengths further. They prove the concept of application of diffractive optical elements with off-axis focusing to non-invasive reflective THz scanning for detection of human skin for cancer.

The article is a further investigation Optics Express 2020, 28, 715–723 where the technique and set-up are well explained. It can suit the journal, nevertheless there is no major innovations in the field.

In terms of specific concerns and/or potential areas for improvement:

  • I suggest a general improvement of almost all the figures:
    • Fig. 1: Improve the caption, e.g. develop and explain the image naming as the "source" or "sample". Explain which type of source or sample (or even the DOE) trying to make the figure self-explanatory without the need to go to the central text.
    • Fig. 2 and Fig. 3: Label all panels in the figure with letters. Better explanation in caption.
    • Fig. 4: Define xd, xs and yd in caption.
    • Fig. 5: Again label all the panels and improve the explanation. Comment in the caption also P values.
    • Fig. 6: Comment in caption P values.
    • Fig. 7: Same as Fig. 2 and 3
    • Fig. 8. Complete the caption, commenting lines and colours of the graph, as it is done in the text.
  • Line 43-45: Develop further by explaining each approach separately
  • Line 53: The authors start the sentence with “another approach” when they are developing the second approach introduced previously. Reformulate the discussion of this paragraph by being clearer and relating with the figure with new labels.
  • Line 127 and related Fig. 6: Pmax in reference PA12 is one, the rest of paraffin have Pmax larger than 1, how is it possible? In Fig. 6 the scale goes from 0 to 1. This does not make sense if there are higher values.

Reviewer 3 Report

The manuscript deals with the design, realization and test of diffractive THz optics. While this is an interesting topic, the manuscript appears more like a lab report than a scientific paper. The authors do describe step by step what they have done and what they have obtained, from calculations as well as from experiments.

However a proper scientific paper should first introduce the topic, then state the open questions addressed; then the paper should describe what the authors did in order to solve these problems and finally give the results.

I do get an idea from the second sentence of the abstract "....we aim at the development of a device for in-reflection examination of samples." But the authors do not show in which way their approach is better than refractive optics. A brief statement somewhere does not give any quantitative improvement. And even more more important, the authors do not mention the use of mirror optics, commonly employed in THz technology.

Here I would expect some statements, although I do understand that a full size quantitative comparison is beyond the scope of this paper.

Round 2

Reviewer 1 Report

In general I am happy with the authors' responses to my points of concern. I would like to thank authors for a well reasoned answers to the points No. 4-6. Also, I think that the manuscript was suitably corrected and now it is much more understandable even to the readers not exact in the field.

Also, concerning point No. 8 I still think that authors should add a sentence something like "XZ scans can be found in..." and to provide the reference to the work they are mentioning there "Siemion, A., et al. 'Terahertz diffractive structures for compact in-reflection inspection setup.' Optics Express 28.1 (2020): 715-723.)".

All in all I think that the manuscript should be accepted for publication after the above mentioned small correction will be addressed.

Author Response

Dear reviewer,

the authors would like to express their gratitude for all the comments that clearly helped us to improve the quality and readability of the manuscript.

According to the suggestion, the following sentence has been added to the article before Fig. 5:
‘The scans of the analogous structures performed along the optical axis can be found in our previous paper [Siemion, A., et al. 'Terahertz diffractive structures for compact in-reflection inspection setup.' Optics Express 28.1 (2020): 715-723.)]. Their character remained the same.'

Best regards,

Paweł Komorowski and the co-authors

Reviewer 3 Report

The authors have addressed most comments and improved the manuscript in this regard. It can be published as it is.

Author Response

The authors would like to thank the reviewer for all the comments that clearly helped us to improve the quality and readability of the manusript.
Best regards,
Paweł Komorowski and the co-authors